# Shame or What Makes Irrational Social Anxiety Rational

**DOI:** 10.3390/healthcare13222891

**Published:** 2025-11-13

**Authors:** Artemiy Leonov, Justin P. Laplante

**Affiliations:** 1Department of Anthropology, Psychology, and Sociology, University of West Georgia, Carrollton, GA 30118, USA; 2Department of Psychological Sciences, University of Connecticut, Storrs, CT 06269, USA; justin.laplante@uconn.edu

**Keywords:** social anxiety disorder, shame, cognitive bias, cognitive-behavior therapy, self-integration

## Abstract

Social anxiety disorder (SAD) is defined as a pathological fear of social interactions in which an individual may be negatively evaluated by others. The crucial component of assessing the state as clinical is the ‘context-insensitivity’ of the fear—i.e., the negative evaluation does not have any tangible repercussions, or the evaluation is not as detrimentally negative, as the patient presumes. However, this model excludes the role of self-conscious emotions, specifically shame, in aggravating social fears. This article models shame as an emotion that is highly aversive, unpredictable, and resistant to metacognitive regulation, which entails a perspective that SAD is the product of high shame-proneness and the inability to voluntarily mitigate it. The evidence of mutual correlates, such as socially prescribed perfectionism and treatment outcomes of cognitive and behavioral therapeutic modalities, is used to justify the argument. The article suggests the possibility of implementing humor as a supplement to a standard SAD-treatment self-regulation strategy that would allow subjects to control the emergence of their shame more efficiently. Finally, a potential randomized control trial study design is proposed to test the perspective outlined.

Social anxiety disorder (SAD) is classified as an intense, overwhelming fear (that is disproportionate to the threat) of one or more social situations in which an individual may be negatively evaluated by the others [1]. Of the mental disorders, SAD is one of the most common and persistent across the lifespan—according to an NIMH review [2], an estimated 12.1% of the US population would have SAD at some point of their lives, and the average duration of the disorder varies from 10 to 29 years [3,4,5]. Living with SAD entails severe impairments in social functioning and long-term goal attainment. People with SAD are less likely to be married, more likely to remain unemployed, and less likely to pursue higher education, as they avoid social situations which may provoke this fear (see [6] for review), which, consequently, negatively affects both their perceived life quality [7] and the socio-economic impact they could have made. For this reason, it is essential for clinical psychologists to devise and imply accurate models of etiology and treatment for SAD. 

Cognitive behavior therapy (CBT) is one of the most researched and effective methods of treating SAD—one of the most recent meta-analyses of after-treatment and long-term effects (12+ months) indicate moderate and lasting reduction in SAD symptoms: Hedges’ *g* = 0.38 for after-treatment assessment and *g* = 0.42 for 12+ months assessment [8]. The present paper aims to suggest a perspective on increasing these effect sizes further by offering a new model of emergence for social anxiety experience based on the self-conscious emotion of shame and discussing the possibilities of its direct regulation. 

## 1. ‘Cognitive Bias’ or ‘Irrational Fear’ Models of Social Anxiety Experience

The most cited models of emergence of social anxiety experience and etiology of SAD are connoted as ‘cognitive’ and present social anxiety as a product of a specific type of attention, stimuli interpretation, or both [9,10,11]. The common tenet of all such models is the idea of ‘cognitive bias’—an attentional or information-processing maladaptation that triggers an “inappropriate” social anxiety response, i.e., “when the danger [to physical or reproductive survival] is more imagined than real” ([9], p. 70). In other words, intense social anxiety response in any mundane situation would be pictured as a maladaptive, i.e., not cost-effective, thus ‘irrational’, atavism.

In addition to the functionalist explanation of the irrationality of anxiety experiences that classifies SAD, the cognitive bias model is supported by intersubjective evidence as well. For example, it is observed that people with SAD typically believe in their own social unattractiveness or skill-related ineptitude and put significant effort into presenting themselves to their potential evaluators, which are imagined as harsh critics [12,13,14]. Contrary to these potential assumptions of people with SAD, in a study by Cartwright–Hatton and colleagues [15], 10–11 years old children with and without SAD had conversations with volunteers who then evaluated the social skills of their interlocutors; they found there was no statistical difference between volunteers’ estimates of social skills of people with and without SAD. Moreover, Taylor and Alden [16], who used the same methodology, found that when people with SAD act ‘naturally’ in conversations, they receive significantly higher estimates than when they actively search for the most appropriate or socially savvy response. Finally, the survey study of Iancu and colleagues [17] concludes that high self-criticism is one of the strongest predictors of Liebowiz Social Anxiety Scale scores (R^2^ = 0.741 stepwise linear regression, controlled for age, gender, and marital status).

Summarizing the results of the studies described, it may be tempting to assume that the intensity of social fears of an SAD-diagnosed person is irrational. Indeed—comorbidity conditions excluded—such people are equipped with the necessary social skills, and others who they interact with are neutral or even forgiving. Therefore, cognitive models would suggest that an effective treatment of SAD would involve restructuring interpretive and attentional biases by building the patient’s confidence in their social skills, teaching them to switch attention from negative cues and self-conscious observations, and helping them to perceive social situations as less self-centered and other individuals as more benign and forgiving [11]. However, there may be a more pertinent approach.

## 2. What Is Feared: Shame

Despite the outlined rationale, it would be a brute simplification to picture social fears as a distorted information processing that prevents the patient from assessing the situation ‘dispassionately.’ The primary counter-argument is derived from the emotion of shame, which is integral to the emergence of the social anxiety experience. Shame, most generally defined, is the emotional state associated with realizing one’s own inferiority, inadequacy, and/or moral corruption, typically under surveillance of an apparent or an imagined observer [18,19,20]. We further demonstrate that if SAD is conceptualized as the fear of experiencing shame rather than the fear of being scrutinized, the ‘irrationality’ of the fear becomes unsound, due to the shame experience being relatively environment-independent and due to the emotional pain that shame entails. Rather, this fear has to be viewed as rational and as an expected response to the possibility of being ashamed, and thus alternate models targeting shame-proneness and shame-reduction could be of benefit.

It is important to discern between shame, guilt, and embarrassment. Guilt is a pro-social emotion that drives apologetic behavior; one ceases to feel guilt when the issue they have caused is perceived to be resolved [21]. Therefore, the treatment of guilt differs from the treatment of shame. For example, the ‘acceptance-based’ cognitive approaches would treat guilt through “self-forgiveness”, i.e., accepting that life can be continued without amending the mistake [22], whereas shame would be treated through “self-acceptance”, i.e., acknowledging that one is an individual worthy of love and understanding [23,24]. The term “embarrassment" is used to describe the state of acute distress from not receiving the expected positive estimate from another person [19]. This state is viewed as a specific (or even primary) component of the complex shame emotion [25], and its treatment is a significant part of anti-shame interventions [24].

The main difference between the fear of shame and the fear of being scrutinized is the specific role of self-consciousness (evaluating the self from the standpoint of a detached observer [25]). While scrutiny presupposes an active involvement of a judicious observer, the shame response can be triggered by the mere assumption of the presence of someone capable of evaluating [26,27,28]. The fear of being ashamed then can partially be theorized as the fear of being merely observed in an unfavorable state though not necessarily punished for it. ’Unfavorable’ may mean “morally corrupt” [26], “incompetent” [27], or “left with rejected excitement” [25,29]. The issue that makes the fear of being ashamed so intense is the malleability of self-standards, i.e., the presumed ‘favorable’ state may rapidly become ‘unfavorable’ when one perceives receiving negative feedback from their evaluator, despite the benign or positive intentions of the latter [25]. The situation is aggravated by the socially prescribed perfectionism that characterizes individuals diagnosed with SAD, i.e., being excessively critical of themselves, ensuring that their appearance or performance matches the social standards and expectations of others, and overly rueing when they realize that they failed to do so [14]. Thus, the emergence of shame becomes extremely unpredictable and uncontrollable, with a risk of being triggered by something as benign as a memory that, while originally not shameful, becomes negatively interpreted in the present context. 

These theses are masterfully illustrated in Fyodor Dostoyevsky’s novel, “The Adolescent” [30]. Someone may argue that literary examples lack the necessary scientific rigor. Therefore, it has to be shown why using literary examples, and Dostoyevsky in particular, is appropriate for bolstering psychological claims. First, the “narrative turn” in psychology has encouraged researchers to approach psychological data, e.g., storytelling, with literary devices, as well as vice versa [31], which further blends the boundaries between ‘real-life legitimate’ and ‘fictional illegitimate’ data for psychological research. Second, some scholars assume that the depiction of neurological and psychiatric conditions in fiction would be useful for specialists to better understand the subjective experience of their patients, which may become crucial for further advancement in treatment and recovery programs [32]. Finally, the works of Dostoyevsky have a reputation among both medical specialists [33] and psychologists [34] of depicting deep and intricate human sentiment in its clearest and most comprehensive fashion. Moreover, there have already been studies which are fully dedicated to the depiction of emotional experience in Dostoyevsky’s novels, for example, the experience of “suffering” [35]. Thus, we stipulate the validity of our literary example as a phenomenological vignette intended at clarifying, illustrating, and analyzing the emotion of shame. 

Now it is possible to describe the example itself. In one of the conversations between Versilov, the noble, and his unlawful son, Arkadii, Arkadii, who has just expressed to Versilov how much he loves him, feels ashamed thinking that his father would decide that his confession was pretentious and tries to assure Versilov that it was genuine. Versilov understands Arkadii’s concerns, and in turn, tells how he confessed to Arkadii’s mother’s husband about their love affair—although his remorse felt genuine to him at the moment, he realized over time that it was inauthentic, and Versilov is ashamed of it, despite no witness ever expressing any doubt in his honesty [30]. 

The empirical evidence for the claim that a significant component of social fears is intrapersonal self-evaluation can be derived from the comparison studies of cognitive and exposure interventions. A recent randomized control trial study performed by de Mooij and colleagues [36] concludes that the cognitive restructuring technique produces a more significant post-treatment effect on the levels of avoidance of social interactions, whereas exposure technique produces a more significant after-treatment improvement in social skills, which ‘evens out’ the general effects on social anxiety mitigation (78% vs. 69% of the sample not meeting the SAD diagnostic criteria post-intervention, respectively). Moreover, the authors observed that the exposure condition would eventually achieve a significant decrease in avoidance of social situations (3-month post-treatment assessment), attributing it to the “developed confidence” in one’s social skills, which the cognitive condition had developed through “restructuring.” In both cases, “confidence" may be interpreted as ‘confidence that the risk of experiencing shame in that social interaction is minimal’ (see Section 3). A more extreme example is Clark and colleagues’ [37] comparison of effectiveness between cognitive and exposure treatments for SAD. In this randomized control trial, participants completed respective 14-week interventions, and at the post-treatment assessment, the percentage of participants who no longer met diagnostic criteria for SAD was almost two times larger for cognitive therapy (86%) than for exposure + applied relaxation (45%). This difference in effect sizes may further support the notion that “social skills” are an intermediary variable in developing a self-perception of a socially competent self, which, in turn, directly reduces social fears. 

Additional indirect evidence may be derived from the etiology of selective mutism (SM)—a child (2–5 age of onset) anxiety disorder that restricts the child’s ability to communicate with people outside their family, despite having the necessary linguistic and cognitive abilities [38,39]. Contrary to other anxiety disorders, fears of SM are not associated with past experience of negative interactions with the object of fear (in case of SM, strangers); moreover, children that are diagnosed with SM typically come from overprotective parents who also have some type of anxiety disorder [40]. Therefore, it may be assumed that the social fears associated with SM are produced by some internal state associated with self-perception rather than environmental feedback. 

## 3. The Problem of Voluntary Mitigation of Shame

It has to be underscored that shame is one of the most aversive emotions a human can experience. Illustrating this aversiveness, sociologists [20], psychoanalysts, [41], and historians [42] note that it has been a culturally normal practice for people to choose physical pain, sacrifice, or even death over anticipated disgrace. Therefore, it is clear why shame-prone patients with SAD would avoid shame by any means. Unfortunately, the strategies people come up with outside psychotherapy, e.g., self-focused attention, safety behaviors, alertness to negative social cues [1], significantly impair their social functioning and openness to social experience. This outcome may be explained by the “two-factor fear theory” [43,44]. Its central notion is that internal states, such as aversive emotions, are learned associatively, whereas response strategies are learned instrumentally, depending on how well they alleviate the negative experience. According to Mowrer [43], being embarrassed in a social situation creates an association of fear with this situation, and according to McAllister and McAllister [44], this fear will drive the most accessible mitigation action which, if successful, will be learned through negative reinforcement. 

The most problematic aspect of shame is its relative resistance to voluntary mitigation. Shame is an emotion borne in human interaction—one that the subject of the shame is participating in, observing, imagining, or remembering [28]—and manifests when a positive or neutral interpretation of an exchange is interrupted by the interpretation of the events as inappropriate, which becomes apparent either when the subject becomes conscious of being observed in an unfavorable state, or when the subject sees their response as unfavorable [25]. Referring to Dostoyevsky’s example above [30], in simpler cases, like Arkadii’s one, shame—more precisely, its ‘embarrassment’ component—would be mitigated by the positive feedback from the observer, e.g., ‘I have not seen anything [or anything inappropriate]’ [26]. In complex cases, like Versilov’s one, where shame is the product of a more mature worldview, it may be especially hard to encourage the patient to see themselves in a non-shameful light, because they have invested reflexive effort into restructuring their self-view, and, as a result, their developed self-judgment is more epistemically complex and accurate [27,45].

Approaching shame regulation from this perspective, it can be seen that cognitive and behavior treatments of social anxiety have not been particularly focused on mitigating shame response itself. Instead, they offer tools for avoiding the generation of potentially shameful memories, e.g., learning to assess the environment instead of the self, thus having no ‘data’ for self-criticism [9] and/or tools for soothing the fear of feeling shame during and after a social interaction, e.g., by applied relaxation techniques that reduce somatic anxiety response [37], or by increasing the patients’ confidence in their abilities to assess their social performance more objectively, thus favorably [46]. As a result, the question remains how the shame should be targeted and regulated in a therapeutic setting. 

## 4. Humor: Integrating the Self as Shame-Regulation

Most shame scholars agree that a distinct feature of the shame response is the ‘split of the self’ into the ‘shaming’ and ‘shameful’ parts [16,19,25,26]. The issue that makes shame so torturous is the stalemate position of a disintegrated self—the ‘shaming self’ is never able to transcend the ‘shameful self’, thus breaking the continuity of the personhood completely and leaving the shameful self relegated to a position of ‘not me.’ Further complicating matters, it is incredibly difficult to be able to accept the shameful self as part of the self, thus restoring the continuity of the personhood.

The issue of self-acceptance is central for the compassion-focused psychotherapeutic approach to the treatment of pathological shame [23,24,47,48]. In this modality, self-acceptance, or “self-kindness,” is encouraged through a specific formula: “treating oneself as one would a loving friend in the midst of his or her pain and suffering” ([48] p. 348). An apparent theoretical issue is that such self-kindness encourages a further separation of the self into the ‘accepting’ and ‘accepted’ parts. Moreover, it is not a self-evident proposition that self-acceptance is practiced the best when shameful memories are accepted “as they are”, i.e., without any additional modification. Addressing the highlighted problem of self-integration, we cannot overlook the instrument of humor. The therapeutic effect of creating witty paradoxes made humor a topic of specific interest for life philosophies and psychoanalysis—theories whose primary concern had always been the explanation of (dis)integration of the world and psyche [49,50,51]. Indeed, as Freud notes, humor allows one to turn ‘inappropriate’ into ‘acceptable’ and ‘private’ into ‘public’—outcomes that should be particularly interesting for the prominent broaden-and-build theory [52], which postulates that substituting a negative emotion with its incompatible positive opposite is the primary source of therapeutic change. More recent clinical scholarship, however, has a limited amount of empirical studies on humor [53] and these studies view humor only as an instrument for a therapist to create rapport and regulate the emotional flow of the session [54,55]. Instead, we assume that humor can be treated as one of the self-regulation tools that a therapist can teach their client to use to overcome shame. For example, Shabad [29], a psychodynamic therapist, reports a case study of treating a shame-prone and socially anxious client, Paul, by teaching him to take a more “playful” approach to himself. Shabad reported using humor to help Paul to cope with Paul’s shame-based rumination about his strict and distant father. The jokes had then encouraged Paul to speak less seriously about himself and "take a chance at reintegrating” (p. 70). Additionally, a qualitative review of narratives of people who have recovered from anorectal illness has shown that humor, particularly, self-irony becomes an important tool to non-judgmentally accept the ‘imperfection’ of the self and be able to communicate about it with others [56]. Finally, it may be reasonable to believe that a person who is able to laugh at their clumsiness in mundane situations is less shame-prone [19], and therefore, should experience less social anxiety. 

While there is little work performed in this area to date, it is a rich area for future investigation to explore the specifics of how, when, and in what ways humor might work as a tool in the toolkit of reducing shame, which will therefore aid in reducing social anxiety. These assumptions could be tested empirically through a randomized control trial study. Such a proposed study could involve four conditions: a standard cognitive therapy for SAD (control group), cognitive therapy + self-acceptance techniques (group 1; controlling for shame), cognitive therapy + humor (group 2), and cognitive therapy + self-acceptance + humor (group 3). The “humor-education” intervention, based on the observations made by Shabad [29] and Yue [56], would involve interacting with the comedic narratives of overcoming social anxiety—thus, controlling for individual psychotherapists’ preferences to make or not to make jokes. Based on the latest RCT findings on integration of mindfulness-based and humor-based approaches to self-acceptance [57], it may be reasonably expected to see the largest effect of social fear reduction in Group 3. A particularly important metric for evaluating the effect would be measures of “social initiative” and "social avoidance” (see [36]) since we assumed that the confidence in shame regulation given by a humorous approach to oneself, as well as its pro-social/interpersonal orientation, would play a significant role in reducing the fear of being ashamed. We would highly recommend future research to take up such an empirical investigation.

## 5. Conclusions

Social anxiety disorder (SAD) is an adverse condition that limits the subjects’ abilities to participate and succeed in social interactions. Viewing the fear of social interactions as the fear of generating potentially shameful memories will allow clinicians to advance their treatment models by implementing methods that will directly target the clients’ sensitivity to the shame response. Considering the essentially self-conscious and self-disintegrating nature of shame, humor becomes an attractive tool for direct self-regulation of shame response. Future studies should explore possibilities of encouraging clients to use humor in their apprehension of social interactions they participate (or have participated) in, thus making the experience appropriate to share with others, which is the opposite of feeling shame.

## Data Availability

No new data were created or analyzed in this study.

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
