# Peer review of "Shame or What Makes Irrational Social Anxiety Rational"

_healthcare, 2025, doi:10.3390/healthcare13222891_

Round 1
Reviewer 1 Report
Comments and Suggestions for Authors
Recommendation
- Decision: Major revisions
- Rationale: Compelling conceptual contribution that could be publishable after
(1) rigorous tightening of claims; (2) addition of a modest “approach” paragraph; (3) balanced integration with CBT literature on shame; (4) clearer clinical guidance, boundaries, and risk management for humor interventionsSection-by-section critique with actionable revisions
Abstract
- “The article suggests a possibility of implying humor...” Replace “implying” with “implementing” or “using.” Avoid implying established efficacy.
- Add one sentence on limitations and status: “This is a conceptual article proposing testable hypotheses; empirical validation is needed.”
- Replace “irrational” framing with “disproportionate” or “context-insensitive” to avoid stigma.
- Specify that humor is proposed as one candidate technique to facilitate self-integration, not a standalone treatment.
Introduction
- Balance: Note that cognitive models include shame-related mechanisms (self-focused attention, self-criticism, negative self-imagery) and that CBT protocols often directly target shame cognitions and behaviors.
- Terminology: Avoid indicating that SAD sufferers’ fears are “irrational” as a blanket statement; couch in terms of “threat appraisal disproportionate to objective danger,” while acknowledging intense subjective cost of shame.
- Consider adding a short subsection positioning shame within existing CBT targets (e.g., behavioral experiments about feared shame outcomes; imagery rescripting for shame memories).
- Tighten claims from the Cartwright-Hatton/Taylor-Alden studies: emphasize that performance may not be objectively poorer and that self-monitoring worsens performance and evaluations.
Section 2.
- Clarify that “rationality” here refers to the subjective cost/utility framing: fear can be “understandable” given shame’s aversiveness, without claiming it is adaptive across contexts.
- Clarify boundaries between shame, guilt, embarrassment and note their different regulatory trajectories in therapy.
Section 3. The Problem of Voluntary Mitigation of Shame
- CBT often targets shame via behavioral experiments, self-compassion practices, perspective-taking, and imagery rescripting. Rephrase “do not approach shame regulation per se” to “have historically emphasized threat appraisal and safety behavior reduction; shame-specific techniques are variably integrated.”
- Briefly mention alternative approaches with explicit shame/identity components (e.g., Compassion-Focused Therapy; emotion-focused therapy elements; imagery rescripting evidence in SAD).
- Suggest operational markers of “voluntary mitigation”: e.g., reductions in self-focused attention under exposure; re-engagement after shame triggers; state/trait measures of shame.
Section 4.
- Specify mechanisms of action with testable hypotheses:
- Cognitive: humor as benign reappraisal, incongruity resolution, perspective-shifting.
- Affective: positive affect broadening; reduction of shame affect intensity.
- Interpersonal: modeling affiliation/acceptance; reducing superiority/inferiority dynamics when used carefully.
- Self-related: integrating “shaming” and “shameful” parts via playful stance toward self-discrepancies.
- propose RCT adjunct designs (CBT ± brief humor module) with outcomes on shame-proneness, self-criticism, self-focused attention, and social functioning; include manipulation checks for humor type (affiliative vs self-defeating).
Section: Literary example and justification
- Frame as phenomenological illustration only; not evidentiary support.
- Consider adding at least one clinical vignette (de-identified) or a composite vignette that shows shame activation and a therapist-guided humor intervention, if journal allows.
Conclusion
- Replace “imply” with “implement” or “apply.”
- Replace “humor becomes an attractive candidate tool…” rather than “the most attractive tool.”
Decision: Major revisions
Author Response
Please see the reply attached.

Reviewer 2 Report
Comments and Suggestions for Authors
Thank you for inviting me to review the manuscript titled “Shame or What Makes Irrational Social Anxiety Rational” for Healthcare. This paper is a conceptual piece that provides a unique perspective on Social Anxiety Disorder (SAD). The authors highlight limitations to the current framework of SAD and offer a possibly new take on the role of shame in the cognitive, emotional, and behavioral features of the disorder. Overall, this manuscript presented a refreshingly unique view of SAD. Below is my feedback for the authors.
- I would recommend the authors expand in their definitional view of SAD in the opening paragraph by including the following: social situations evoke or provoke fear (or anxious) responses, social situations are avoided, and the fear is “out of proportion to the threat.”
- The authors refer to the effect sizes as “scores” (page 1, line 40). I suggest they use the term “effect sizes” instead.
- The authors should consider briefly summarizing the two-factor theory of anxiety – initial fear is explained by classical conditioning, ongoing avoidance and escape behavior is explained by operant conditioning. See McAllister and McAllister (1995) for a thorough review.
- Page 2, Line 53: The authors use the phrase “functional explanation” when introducing the irrationality of anxious experiences. I would recommend avoiding this wording, as it implies a casual explanation.
- Page 2, Lines 65-66: I recommend the authors expand on their point that high self-criticism strongly predicts social anxiety. Where is the evidence for this? What empirical research supports this statement.
- Page 2, Lines 67-68: Similarly, the authors should provide evidence for the statement that SAD-prone individuals may have better emphatic skills than those less prone to social anxiety. Also, what are “emphatic skills”?
- The authors make some great points in Section 2. What is Feared: Shame.
- Page 3, Line 92: I recommend defining “self-consciousness” and highlighting some of the limitations of the term from a methodological perspective (e.g., difficult to observe).
- Page 3, Lines 95-96: What is meant by the term “unfavorable state”?
- Page 3, Lines 131-137: It might be a leap to use evidence of cognitive therapy’s effectiveness to suggest that self-consciousness plays a role in the development of social anxiety. I recommend the authors consider another avenue to support their assertion.
- Page 3, Lines 131-137: The authors cite one study (i.e., Clark et al., 2006) when claiming cognitive therapy is more effective at treating social anxiety than exposure therapy (plus applied relaxation). Yet, there are other studies that offer a different conclusion – that exposure-based interventions are more effective than cognitive restructuring (a component of cognitive therapy) when treating social anxiety (e.g., de Moouj et al., 2023). Given this point and the feedback in item 10 above, the authors should consider revising this paragraph so more emphatic support might be provided for the claim that self-consciousness is important in the conceptualization of the emergence of fear in SAD.
- Page 4, Lines 165-166: Cognitive restructuring and helping clients address irrational fears by exposing them to their cognitive biases or faulty heuristics might address “shame regulation.” Consider revising.
- Conclusion: In several ways, this paper presents a novel way to conceptualize SAD by incorporating I appreciate the authors mentioning future studies exploring how to help clients explore the use of humor in their apprehension of social interactions. However, I think the directions for future research section should be expanded. I would include the following as directions for future research: More research exploring the connection between shame and SAD and additional research on integrating this new conceptualization about shame and SAD into existing CBT frameworks.
Author Response
Please see the reply attached

Round 2
Reviewer 1 Report
Comments and Suggestions for Authors
The paper has been edited according to the edits suggested. The paper can be accepted
Reviewer 2 Report
Comments and Suggestions for Authors
The authors' edits improve the manuscript significantly. I recommend the editorial team publish the paper in its present form.